# Atlantic tropical cyclones downscaled from climate reanalyses show increasing activity over past 150 years

Kerry Emanuel [1]✉

Historical records of Atlantic hurricane activity, extending back to 1851, show increasing activity over time, but much or all of this trend has been attributed to lack of observations in the early portion of the record. Here we use a tropical cyclone downscaling model driven by three global climate analyses that are based mostly on sea surface temperature and surface pressure data. The results support earlier statistically-based inferences that storms were undercounted in the 19th century, but in contrast to earlier work, show increasing tropical cyclone activity through the period, interrupted by a prominent hurricane drought in the 1970s and 80 s that we attribute to anthropogenic aerosols. In agreement with earlier work, we show that most of the variability of North Atlantic tropical cyclone activity over the last century was directly related to regional rather than global climate change. Most metrics of tropical cyclones downscaled over all the tropics show weak and/or insignificant trends over the last century, illustrating the special nature of North Atlantic tropical cyclone climatology.

---

[1] Lorenz Center, Massachusetts Institute of Technology, 77 Mass. Ave., Rm 54-1814, Cambridge, MA 02139, USA. ✉email: emanuel@mit.edu

Changes in the incidence of tropical cyclones are among the most important consequences of climate change, but the short length and often poor quality of historical records of tropical cyclone activity impede detection of climate signals.

Only ~12% of the world's tropical cyclones occur in the North Atlantic, but owing to their closer proximity to populated land and to routine airborne reconnaissance since the mid-1940s, they are better observed than tropical cyclones elsewhere[1]. Before the era of globally comprehensive satellite coverage, generally regarded as commencing around 1980, observations of tropical cyclones outside the Atlantic were insufficient to achieve complete detection, and estimates of storm intensity were few and subject to large error[1]. (An exception was the western North Pacific region, in which some cyclones were surveilled by aircraft from 1945 to 1987.)

Virtually all known reconstructions of tropical cyclone characteristics over the globe have been compiled in a public database known as the International Best Track Archive for Climate Stewardship (IBTrACS)[2]. Outside the Atlantic and western North Pacific, this record is sparse before the satellite era; for example, no tropical cyclone wind speed observations were recorded in the southern hemisphere prior to 1956. In general, the observations become increasingly sparse going back in time.

The IBTrACS North Atlantic tropical cyclone record dates to 1851. Before routine airborne reconnaissance began in the mid-1940s, this record is based on observations from ships and land, including islands. While much of the U.S. eastern seaboard and the Caribbean region was densely populated during this whole period[3], both the number and spatial distribution of ship observations evolved in response to changing demand for goods, sizes of ships, and the opening of the Suez Canal in 1869 and the Panama Canal in 1914[3]. In addition, the increasing use of the "Law of Storms" developed in the early 19th century by Reid[4], Redfield[5], and others enabled ships to avoid the cores of tropical cyclones thus affecting observations of the precise location and magnitude of the storms.

Beginning with the work of Vecchi and Knutson[6] and continuing through a recent paper by Vecchi et al.[7], attempts have been made to correct for missing tropical cyclones early in the record by estimating how many contemporary (typically post-1971) cyclone tracks would have been observed in the pre-satellite era from land or from ships, with the latter estimates based on the statistics of digitally available ship locations going back to 1851. Such analyses must make a number of assumptions, including thresholds for an observation to count as a tropical cyclone record, the distribution of winds with respect to the storm center, assumptions about whether high winds affecting land would have indeed been recorded, and estimates of whether a succession of reports represent a single storm or multiple storms. The method does not account for possible shifts in the geographic distribution of storms or for changing strategies of ships to avoid the cyclones. The method also privileges the null hypothesis and would diminish the magnitude of any real trend that was present[6]. With these assumptions in mind, this body of research concludes that there are no detectable trends in hurricanes or major hurricanes (Saffir-Simpson category 3 and above) through the entire record, from 1851 to various times in the 21st century.

It is important to distinguish between the lack of an actual trend and the lack of a detectable trend. In essence, the aforementioned work demonstrates that the historical data do not suffice to reject the null hypothesis of no trend in hurricanes or major hurricanes. Any number of violations of the stated assumptions could result in an actual trend, but there would be no way to detect such a trend using direct historical observations of tropical storms and reasonable, conservative assumptions

about the null hypothesis and the detectability of the storms with the observational technologies and platforms of the past.

Here we present results of an alternative approach to estimating past hurricane activity, based on dynamical downscaling of three climate reanalyses, spanning more than a century and assimilating only surface pressure, sea ice, ocean surface temperature observations, and in one case, marine surface winds. The downscaled North Atlantic tropical cyclone metrics show consistent and substantial increases over the periods of the reanalyses, interrupted by a prominent hurricane drought in the 1970s and 80s, in agreement with uncorrected historical records. Yet downscaled tropical cyclone activity over all the tropics shows no trend in overall frequency but small increases in the more intense storms.

## Results and discussion

For the present work, I downscaled three analyses: The NOAA 20th Century reanalysis, version 2c[8] (1851-2014) and version 3[9] (1836-2015), and the European Centre for Medium-Range Forecasts CERA-20C renanalysis[10] (1901–2010). The CERA-20C reanalysis differs from the NOAA reanalyses in assimilating marine surface winds and assimilating data into a coupled ocean-atmosphere model. I created 100 North Atlantic and 100 global tropical cyclone tracks for each year in each reanalysis, recording the frequency as proportional to the ratio of successful to net (failed plus successful) seeds, as in previously published work[11–13]. Details of these reanalyses and the downscaling technique are described in the Methods section.

Atlantic tropical cyclone activity is expressed here by three metrics: the total number of landfalling events, the number of hurricanes in the basin and the number of major hurricanes in the basin. Here we define landfalling events as the passage of the storm center from sea to land, based on a ¼ degree bathymetry/ topography data set, and only include storms whose lifetime maximum intensity exceeded 40 kts (21 ms$^{-1}$). Because, in the early part of the observational record, it is not always clear whether successive landfalls were by one or more storms, each landfall was here counted as an "event" regardless of whether one or more storms were involved. As this definition is applied to both synthetic and observed storms, the comparison is homogeneous. Note that all landfall events are included, not just landfalls in the continental U.S. During the period considered here, not just the U.S. but much of the Caribbean was densely populated with adequate sources of information about storms[3].

The other two metrics are the total number of hurricanes and major hurricanes in the North Atlantic basin, with hurricanes and major hurricanes defined as tropical cyclones with lifetime maximum winds exceeding 63 kts (32 ms$^{-1}$) and 95 kts (49 ms$^{-1}$), respectively.

**Trends and variability in dynamically downscaled tropical cyclones.** Figure 1 shows the 7-year running means of these quantities downscaled from all three reanalyses together with their historically observed counterparts as recorded in the (uncorrected) IBTrACS data. The choice of 7 years was motivated by a desire to minimize the purely random variability that dominates especially the historical time series at high frequencies, while retaining most of the variability at decadal and longer periods.

The red curves show the downscaled results, which can be compared to the (uncorrected) historically derived quantities (blue curves). All the downscaled variables have been multiplicatively rescaled to match the mean historical data in the last 50 years of the record. Poisson regression curves are also displayed, and the shading is a measure of the sampling error of

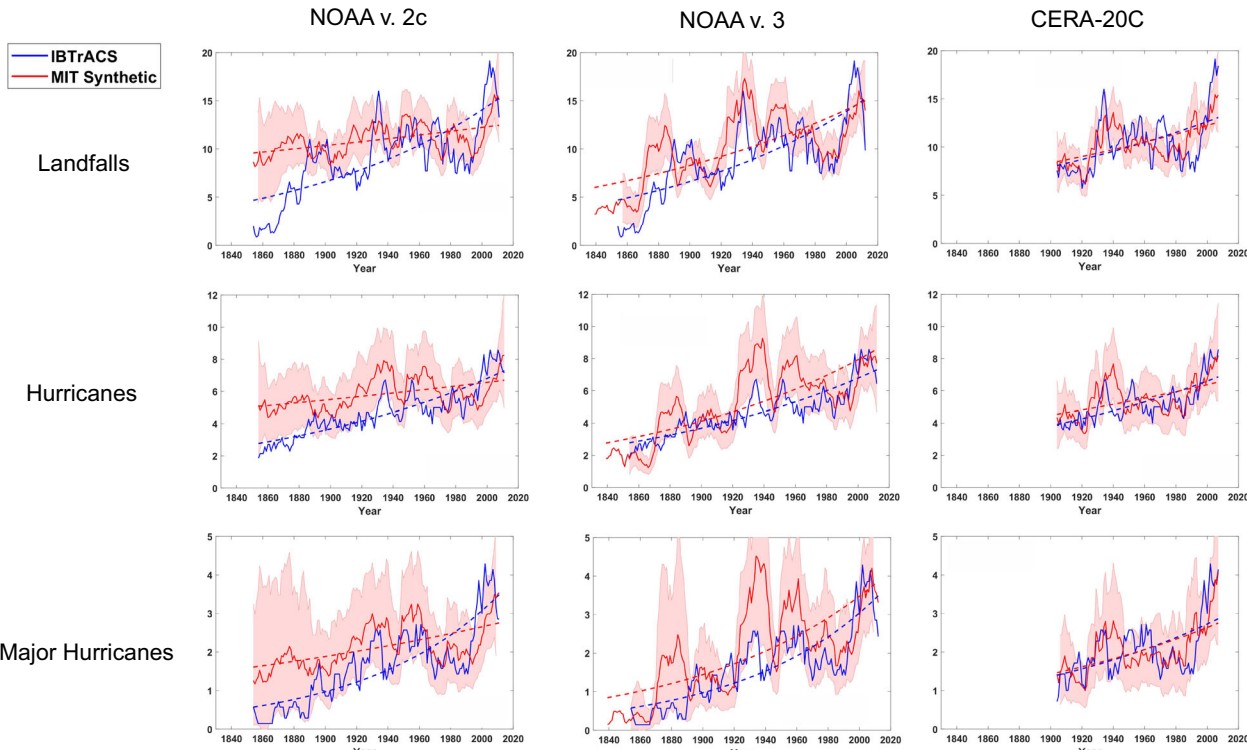

**Fig. 1 Time series of North Atlantic landfall events, basin hurricanes, and major hurricanes.** The landfall frequency (top row) includes only storms whose lifetime maximum winds speeds exceed 40 kts (21 ms⁻¹). Middle row shows basin frequencies and bottom row shows basin-wide major hurricane frequency. Red curves show quantities associated with tropical cyclones downscaled from three global climate reanalyses (left to right) and blue curves are derived from uncorrected historical tropical cyclone (IBTrACS) data. All quantities have been smoothed with a 7-year running mean. The downscaled counts have been multiplicatively rescaled to match the mean of the last 50 years of the historical data. The dashed curves show the Poisson regressions of the downscaled and historical data. The shading around the downscaled counts shows the bounds within which 90% of subsamples of the downscaled data lie, with the subsample size equal to the historical counts.

the historical data. (Hereafter we use Poisson regression for integer variables and linear regression otherwise.)

All of the regression curves shown in Fig. 1 have small $p$-values, with the largest (0.004) associated with the downscaled landfall counts from the NOAA v.2c reanalysis. This indicates that all of the downscaled Atlantic metrics shown here have statistically significant upward trends over the period, and the smallest trends are those of the tropical cyclones downscaled from the NOAA v.2c reanalysis (the oldest of the three). The pronounced upward trends in these metrics stand in contrast to historical records augmented with estimated missing storms[6,7,14].

Storms downscaled from the most recent two reanalyses show very good agreement with the historical data back to 1900, but tropical cyclones downscaled prior to 1900 generally show larger values than those of the historical record, likely reflecting missing storms in the earlier historical record[7]. Poisson regression of the downscaled landfall counts against the historical metrics during the period of mutual overlap of all three reanalysis (1901–2010) yields $p$-values of 0.09, $3.8 \times 10^{-6}$ and $2.5 \times 10^{-5}$ for NOAA v.2c, NOAA v.3, and CERA-20C, respectively.

Downscaled and historical frequencies of tropical cyclones crossing the coastline of the continental U.S. are shown in the top row of Fig. 2. As before, only storms whose lifetime maximum wind speed exceeds 40 kts (21 ms⁻¹) are included, but in this case each storm is counted only once.

Both the observed and all of the downscaled U.S. landfall frequencies show upward linear trends that are statistically significant at the $p = 0.05$ level. The downscaled frequency trends during the 20th and 21st centuries are small but still significant, yet the observed trend during this period is neither positive nor

statistically significant, in agreement with previous work that showed little change in continental U.S. landfall counts[15,16].

All three downscalings show statistically significant increases in tropical cyclone power dissipation index at landfall. The power dissipation index is the sum over each year of the cube of the maximum wind speed in each storm at the time of landfall, and it is a measure of the total dissipation of kinetic energy at landfall, not accounting for the effect of storm diameter[17]. Power dissipation is also a loose measure of the destructive potential of wind storms[17]. These increases are also highly statistically significant over just the 20th and 21st centuries, signifying increasing destructive potential.

Seven-year running means of downscaled tropical cyclone frequencies and major hurricane counts are shown for the whole globe in Fig. 3. Here we show the observed quantities from 1980 on, as they are unreliable prior to then. Each downscaled set has been multiplicatively scaled so that the last 40 years match the mean of the observed quantities between 1990 and 2010. As with the Atlantic, only tropical cyclones whose lifetime maximum wind speeds exceed 40 kts (21 ms⁻¹) are included. Tropical cyclone frequencies downscaled from the two NOAA reanalyses show statistically significant trends, one upward and one downward, while those downscaled from CERA-20C show no significant trend. Major hurricanes downscaled from the two NOAA reanalyses show statistically significant upward trends, but most of these trends occurred before 1920 and the major hurricanes downscaled from CERA-20C show no significant trend.

To help understand how much of the variability and trends in these global quantities reflect actual climate signals, a second

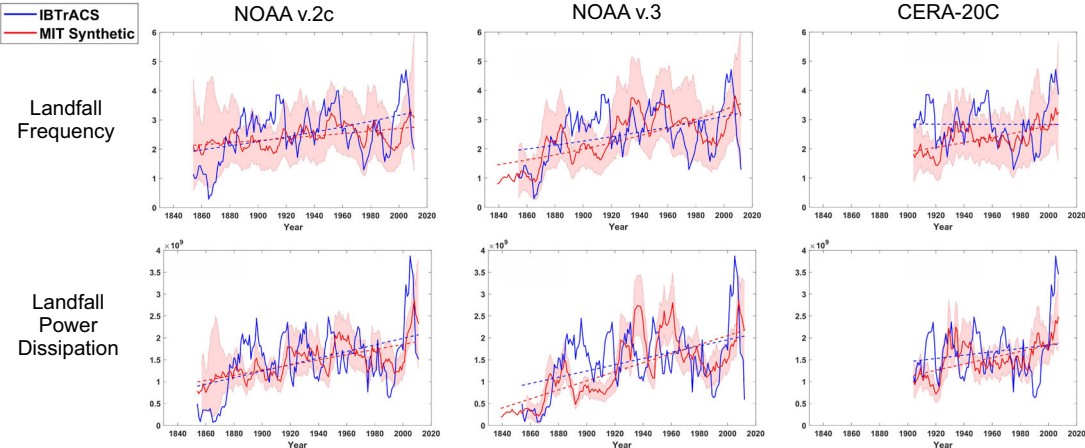

**Fig. 2 As in Fig. 1 but showing the continental U.S. annual landfall frequency and U.S. landfall power dissipation index.** The landfall frequency (top row) includes only storms whose lifetime maximum wind speeds exceed 40 kts (21 ms$^{-1}$). The landfall power dissipation index (bottom row) is in units of m$^3$s$^{-3}$. The linear regression lines in the bottom row are from ordinary linear regression.

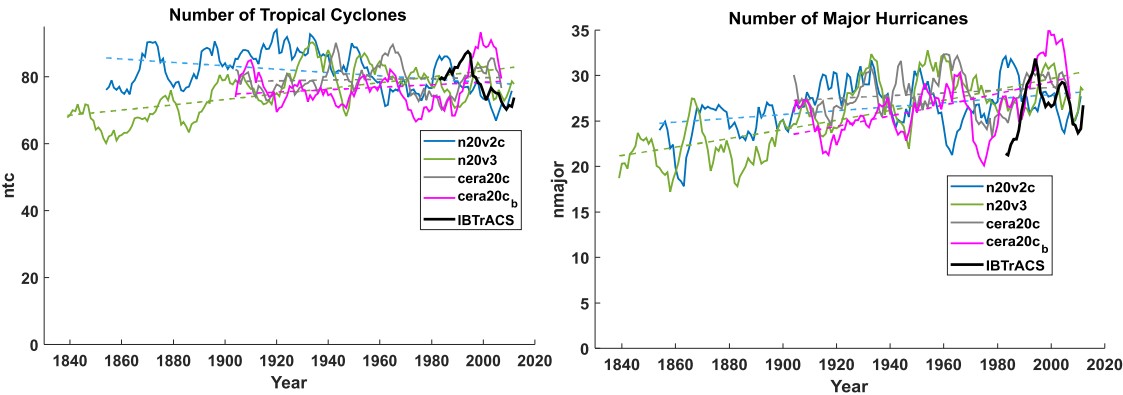

**Fig. 3 Global annual numbers of tropical cyclones and major hurricanes from the IBTrACS historical data and downscaled from the three reanalyses.** Global tropical cyclone count shown at left and major hurricanes at right; reanalyses used are indicated in the legends. Downscaling from a second ensemble member of CERA-20c is also shown here by the magenta curves, and the historical data are shown by the thicker black curves. Dashed curves show the Poisson regressions for each reanalysis downscaled.

ensemble member of CERA-20C was also downscaled; the results are shown by the magenta curves in the figures. This captures both the different variability of the global reanalysis climate simulated by the second ensemble member and a different random variability owing to the limited number (100) of tracks per year. Both the variability and the trends of the second ensemble member differ from the first by order unity, suggesting that neither is statistically robust when the random variability in the reanalyses is accounted for.

The net picture is one of little change over the period. If there are actual climate trends in any of these global metrics, one would have to produce many more tracks per year to reduce the magnitude of the strictly random variability and to run several ensemble members of the reanalyses.

**Comparison of the North Atlantic and global downscaled tropical cyclone climatologies.** Tropical cyclones downscaled from three climate reanalyses show unequivocal increases in Atlantic tropical cyclone activity. Curiously, the same downscaling shows little systematic change in global tropical cyclone metrics. What can account for the large differences between the behavior of downscaled storms in the North Atlantic and elsewhere? Is it possible that defects common to all three reanalyses and/or to the downscaling technique are responsible for these differences?

One possible source of bias in downscaled tropical cyclone trends is the assimilation of historical tropical cyclone tracks into the NOAA reanalyses. (Although, nominally, these data were also assimilated into CERA-20C, in practice many of these observations were rejected by the data assimilation system[9]). As we have discussed, tropical cyclone observations have evolved over time, and it is clear, for example, that there are missing storms in the North Atlantic in the late 19th and perhaps early 20th centuries. Over the rest of the globe, tropical cyclone observations that include intensity data are sparse prior to the era of global satellite coverage that commenced around 1980 (except for the western North Pacific where airborne reconnaissance was conducted from 1945 to 1987). Assimilating tropical cyclone data into these reanalyses potentially introduces artificial trends.

The downscaling model uses only monthly mean thermo-dynamic inputs on the relatively coarse reanalysis grids, thus the input data have no individual tropical cyclones but may be biased by their collective presence. In general, one may expect the assimilation of tropical cyclones to slightly lower the regional monthly mean surface pressures, and raise the temperature and lower the humidity of the troposphere. (All forms of aggregated convection, while they moisten the immediate vicinity of the aggregates, dry the troposphere as a whole[18,19]). Lowering ambient surface pressure causes an increase in tropical cyclone potential intensity[20], which is one of the main inputs to the

downscaling model, but a constant surface pressure was used in calculating potential intensity from the two NOAA reanalyses so in those cases the surface pressure trend would not directly affect the downscaling. Increasing temperature and decreasing humidity in the troposphere would lead to a decrease in potential intensity and would favor decreased genesis frequency; thus, one would expect artificially increasing historical tropical cyclones assimilated into the reanalysis to cause a spurious negative trend in downscaled storms. Moreover, the rather abrupt increase in assimilated tropical cyclone data for the world outside the North Atlantic at the dawn of the era of satellite observations would produce a comparably abrupt change in downscaled tropical cyclone metrics if indeed the inputs to the downscaling were appreciably affected by the tropical cyclone data. No such abrupt changes ~1980 are apparent in Fig. 3. For these reasons, we do not believe that the assimilation of historical tropical cyclone data into the reanalyses is an appreciable source of bias in the downscaled tropical cyclones.

Artificial trends in downscaled tropical cyclone metrics could be induced by biased trends in surface pressure and/or sea surface temperature assimilated into the reanalyses, resulting from the large increase in the number of observations and/or changing instrumentation. With regard to the latter, one would need to explain why such bias trends should be peculiar to the North Atlantic, given that changes in ship instrumentation should have affected observations everywhere. To explore how increasing density of marine observations might have affected the reanalyses, we calculated the surface pressure in the NOAA v.3 reanalyses averaged over each of five tropical cyclone genesis regions and over their respective hurricane seasons. The five basins are the North Atlantic, eastern North Pacific, western North Pacific, North Indian Ocean, and the southern hemisphere. In all of these regions, there are statistically significant upward trends in surface pressure over the period 1836–2015, amounting to 1–2 hPa over the period. The global average surface pressure contributed by all atmospheric constituents except water vapor is held constant in NOAA v.3, and the increase in surface pressure owing to increasing water vapor, assuming fixed relative humidity, is about an-order-of-magnitude less, implying that there is a decrease in surface pressure outside the hurricane genesis regions; this is indeed the case in the NOAA v.3 reanalysis[9].

For the sake of argument, suppose that all of the increase in surface pressure in tropical cyclone genesis regions is an artifact of increasing observations. Further suppose that this increase is compensated hydrostatically by a decrease in tropospheric temperature, constraining the latter to adhere to a moist adiabatic profile. Simple thermodynamics demonstrates that this would lead to an artificial increase in tropical cyclone potential intensity of $\sim 1\,\mathrm{ms}^{-1}\,\mathrm{hPa}^{-1}$, thus the ~1.2 hPa artificial increase in surface pressure in the North Atlantic genesis region would lead to an artificial increase in potential intensity of $\sim 1.2\,\mathrm{ms}^{-1}$, which can be compared to the actual increase of $\sim 6\,\mathrm{ms}^{-1}$ in the reanalysis potential intensity in the same region at the same time of year.

To this we add that the increase in genesis region surface pressure in the other tropical cyclone regions, if they are also artificial, would yield about the same artificial increase in potential intensity, yet there are no significant increases in downscaled tropical cyclone activity in these other regions.

**Comparison to statistically corrected North Atlantic tropical cyclone records.** Tropical cyclones downscaled from three climate reanalyses show unequivocal increases in Atlantic tropical cyclone activity, in contrast to previously published work based directly on tropical cyclone observations and which attempts to correct for missing storms by sampling contemporary tropical

cyclones using statistics of digitally available historical ship routes[7,14]. These two very different approaches agree, however, that the historical record prior to 1900 likely misses some events, and that there has been little change in continental U.S. tropical cyclone frequency over the 170-year record. The downscaling does suggest, however, that U.S. landfalling storms have become more powerful over time.

Aside from possible biases in the downscaled climatologies, it is possible that the statistical corrections that have been applied to the historical data to account for missing storms have been too large. This could happen, for example, if the various detection thresholds used to determine whether a given storm would have been observed by a given ship or land station were too severe, or if there has been a shift of Atlantic tropical cyclone activity toward regions that were previously more data sparse. Moreover, the method of estimating the number of missing storms privileges the null hypothesis of no trend, leading necessarily to underestimates of any real trends that exist[14]. It should be remarked that the ship data recorded in the ICOADS data set used in reference[7] are not equivalent to those used in reconstructing historical hurricane tracks. For the earlier periods, those who produced the historical tropical cyclone archive used, in addition to ships logs, general news and marine intelligence published in major newspapers[21]. This may include some reports from ships whose log books have not yet been digitized and are therefore absent from the ICOADS data; indeed, the digitization of older ship observations is a vigorous, ongoing activity with several updates beyond the dataset used by reference[7] to estimate missing marine observations of tropical cyclones[22].

**On the causes of North Atlantic tropical cyclone variability and trends.** Let us suppose, for the sake of argument, that the downscaling presented here is broadly correct. There are four features of the downscaled Atlantic tropical cyclone climatology shown in Fig. 1 that stand out:

(1)  A substantial upward trend in most metrics
(2)  A local maximum in most metrics in the 1930s and early 40s
(3)  A profound depression of activity in the 1970s and 80s
(4)  A pronounced uptick in activity after 1990

With the exception of the upward trend, these features are also present in the statistically corrected historical record[7].

What aspect of the physical climate evolution of the North Atlantic region could possibly explain these features, which are not present in the global downscaled tropical cyclone climatology?

On clue comes from a genesis potential index (GPI) calculated over the tropical North Atlantic and over all of the tropics. These indices represent empirically determined relationships between space-time distributions of observed tropical cyclones and monthly mean variables such as potential intensity, ambient low-level vorticity, tropospheric humidity, and wind shear, determined from reanalysis data. We here use a particular GPI[23] which has been found to correlate well with rates of genesis of tropical cyclones downscaled using the technique described in this paper. This index is well correlated with rates of genesis of storms downscaled from all three reanalyses, both for the Atlantic and for all of the tropics, with the single exception being the NOAA v.2c for the globe. (For NOAA v.2c, NOAA v.3, and CERA-20C, respectively, the Atlantic correlation coefficients are 0.84, 0.94, and 0.93, and the global coefficients are 0.24, 0.87, and 0.76.)

It is clear from this analysis that the main contribution to increasing Atlantic tropical cyclone activity comes from

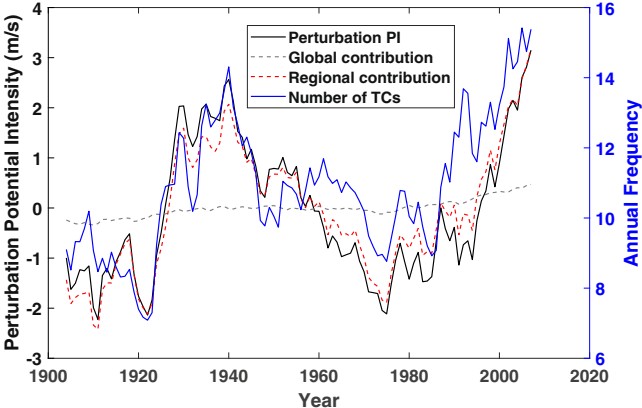

**Fig. 4 Comparison of Atlantic tropical cyclone frequency to potential intensity anomaly and decomposition of potential intensity into global and regional contributions.** Seven-year running means of Atlantic Main Development region (6°–18° N, 20°–60° W) August–October departures of potential intensity (black, ms$^{-1}$) from its mean over the record, and its decomposition into global (dashed green) and regional (dashed red) components. The 7-year running mean of the annual frequency of tropical cyclones downscaled from CERA-20C is shown in blue (right axis).

increasing potential intensity, a measure of the thermodynamic potential for tropical cyclones. (Annual values of late summer Atlantic GPI and potential intensity are correlated with an r$^2$ of 0.83.) Recently, a technique has been developed to linearly decompose changes in potential intensity into global and regional contributions, with the global contribution assumed to result directly from global climate change[24]. Figure 4 shows 7-year running means of potential intensity averaged over August–October and over the Atlantic "Main Development Region" (6–18 N, 20–60 W) together with its decomposition into global and regional contributions. The mean over the record has been subtracted from the potential intensity. (The sum of the two terms is very nearly, but not exactly, equal to the full perturbation potential intensity.) These are compared to the annual frequency of tropical cyclones downscaled from the CERA-20C reanalysis. This clearly demonstrates the near total dominance of regional climate variability over global climate change in controlling downscaled Atlantic tropical cyclone variability, as was also concluded by a recent study[7]. The direct contribution of global climate change to potential intensity is not entirely negligible, with a highly statistically significant increase of about 0.9 ms$^{-1}$ over the period; nor can one rule out an indirect contribution of global change to some of the North Atlantic variability. For example, global warming can lead to changes in the meridional overturning circulation of the ocean, which has a strong effect on North Atlantic climate[25].

A prominent feature of the observational record of North Atlantic tropical cyclones, storms downscaled from all three reanalyses, as well as tropical Atlantic sea surface temperature and potential intensity, is the pronounced decline from World War II to the late 1970s followed by a rapid rise to the present. This was originally linked to anthropogenic sulfate aerosol emissions by Mann and Emanuel[26] and subsequent modeling work[27,28] supports such a link, but recent research[29] strongly suggests that the radiative response to enhanced mineral dust over the tropical North Atlantic, resulting from a sulfate-aerosol-induced drought in the Sahel, is responsible for up to half the decline in sea surface temperature and Atlantic hurricanes during this period. The decline and subsequent recovery of Atlantic tropical cyclones closely follow the large post-war increase in sulfate emissions, followed by an equally strong decline as a result

of the implementation of clean air regulations. A previously argued role for a naturally occurring multi-decadal climate mode known as the Atlantic Multi-decadal Oscillation[30] has been recently cast into doubt[31].

Another interesting feature present in all three downscaled Atlantic tropical cyclone records and the historical record is the strong increase in activity during the 1920s. This is also reflected in tropical North Atlantic sea surface temperatures and related quantities such as potential intensity. The increase led to a local peak in activity in the 1930s, corresponding to the North American Dust Bowl period when continental temperatures were also high[32].

The centennial changes in North Atlantic regional climate, including tropical cyclone activity, remain an interesting and important topic of research in climate science, and the quality of predictions of future cyclone activity will depend on improved understanding of the causes of recent variability. And, while global measures of tropical cyclone numbers and wind intensity show little change in these results, increasing temperatures strongly imply increasing tropical cyclone rainfall[33]; a leading cause of damage and mortality from these storms[34]. Moreover, applying the same downscaling technique used here to global climate models indicates substantially increasing tropical cyclone activity in response to global warming[12,13], particularly in the northern hemisphere. Clearly the interaction of tropical cyclones with regional and global climate remains a fruitful and important topic of scientific research.

## Methods

**Climate reanalyses.** Three 20th century reanalyses were employed in this study: NOAA 20th Century reanalysis, version 2c[8] (1851-2014) and version 3[9] (1836-2015), and the European Centre for Medium-Range Forecasts CERA-20C renanalysis[10] (1901-2010). These reanalyses assimilate actual observations into global weather forecast models to arrive at optimal historical state estimates of the atmosphere and sea surface. Unlike conventional reanalyses, though, 20th century reanalyses are constrained only by surface pressure, sea ice, ocean surface temperature observations, and in the case of CERA-20C, marine surface winds. These observations are considered good enough over the past 170 years or so to provide high-quality global climate analyses when assimilated into contemporary numerical weather forecast models[8–10]. The reanalyses also account for changing greenhouse gas concentrations, natural and anthropogenic aerosols, and solar variability. Their horizontal resolution is too coarse to simulate tropical cyclones with any fidelity, but they do assimilate historical records of tropical cyclones, such as they are. We do not here make use of any tropical cyclones explicitly represented in these reanalyses, and indeed their inhomogeneous presence in the reanalyses is an impediment to interpreting the results, as was discussed in the main body of this paper.

One of the most important differences among these reanalyses is the sea surface temperature (SST) data set assimilated into them. Specifically, prior to 2013, the NOAA v.2c reanalysis assimilates Sparse Input (SODAsi) version 2 (SODAsi.2[35]) with the high latitudes (>60) corrected to COBE-SST2[36]. In the NOAA v.3 reanalysis, the SST data were updated to SODAsi.3[37]. Although the CERA 20c reanalysis used an interactive ocean, its surface temperature was relaxed back to SST data using the HadISST2 data set[38]. Chan et al.[39] demonstrated that different historical gridded SST data sets can appreciably affect explicitly simulated tropical cyclones in global models, and these different SST reconstructions no doubt contribute to the differences in downscaled tropical cyclone activity.

**Dynamical downscaling.** The technique for downscaling tropical cyclones from coarse-resolution global climate models and reanalyses is described in detail in refs. [11,40]. For each year in the dataset, monthly mean sea surface temperature and atmospheric temperature and humidity are used to calculate monthly mean potential intensity and 600 hPa specific humidity and temperature, all of which are inputs to the intensity model described below. The potential intensity was calculated using the algorithm described in Bister and Emanuel[41]. Daily horizontal winds at 850 and 250 hPa were processed to calculate the monthly means, variances, and covariances among both components at both levels. These were then used to synthesize time series of winds at both levels, constructed as Fourier series of random phases in time constrained to have the correct monthly means, variances, and covariances and to obey the geostrophic turbulence kinetic energy spectrum in which the kinetic energy per unit wavenumber falls of as the inverse cube of the wavenumber (see supplement to Emanuel et al.[40]).

The time-evolving climate state is then seeded, randomly in space and time, with weak proto-vortices that serve to initialize the intensity model. The random

seeding tapers off toward the equator to prevent storms forming there. The vortices move with a weighted average of synthesized winds at the 250 and 850 hPa levels, according to a "beta-and-advection" model[42]. Their intensity is calculated using a deterministic, coupled ocean-atmosphere quasi-balanced axisymmetric numerical model with parameterized effects of environmental wind shear[43], which is supplied from the synthetic wind time series. The wind and thermodynamic fields are interpolated linearly in time to the given date and time, and bi-linearly in space to the given latitude and longitude of the storm center.

If the intensity model fails to predict an intensity >7 ms$^{-1}$ within 2 days of initiation, the seed is discarded. In practice, the vast majority of seeds fail this test; only those placed in suitable thermodynamic and kinematic environments thrive. The process is therefore somewhat akin to natural selection. For the present project, we also required the synthesized tropical cyclones to reach a maximum wind speed of at least 40 kts (~20 ms$^{-1}$) during their lifetimes. The numbers of both successful and unsuccessful seeds are recorded, and the process continues in a given year until the preset number of successful storms has occurred. The frequency of storms in the given year is taken to be proportional to the ratio of successful to unsuccessful seeds, with the single proportionality constant determined by comparing the time mean to historical observations over the period in which they are deemed reliable.

This method has been shown to reproduce many important facets of observed tropical cyclone variability, including histograms of lifetime maximum winds, basin counts, the annual cycle in each basin, the global spatial distribution genesis and track density, interannual variability in the Atlantic, and decadal trends in power dissipation in each basin except the eastern North Pacific[11]. It also captures the observed poleward migration of the latitude of maximum intensity of tropical cyclones in the western North Pacific[44], and statistically clusters in a way that is consistent with observations[45].

## Data availability

The downscaled hurricane track data and data analysis software are freely available for research and education purposes only and may be obtained by contacting the author at emanuel@mit.edu. Recipients will be asked to sign a non-redistribution agreement.

## Code availability

The data contained in all 4 figures as well as MATLAB scripts for plotting the data are freely available at https://zenodo.org/record/5558992.

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

## Acknowledgements

This research was supported by the National Science Foundation under grant ICER-1854929. The author is grateful to Gil Compo of NOAA for his support and advice.

## Author contributions
K.E. designed and performed the simulations, carried out the analysis, and wrote the paper.

## Competing interests
The author declares no competing interests.
