## [Peer Review File · Nature Communications]

REVIEWER COMMENTS

Reviewer #1 (Remarks to the Author):

Review for "Atlantic tropical cyclone activity downscaled from climate reanalyses shows increasing risk through the late 19th and 20th centuries,"

By Kerry Emanuel

Summary: In this short paper, Prof. Emanuel applies his statistical downscaling technique to try to estimate changes in hurricane/TC activity over the last 150 years. The downscaling technique simulates the growth and decay of very idealized TCs with many randomly generated regional wind fields generated to have mean and statistical properties of temperature, humidity, and weather produced in global models. In this case the global fields come from 3 "Climate Reanalyses" which are forced by surface pressure and SST measurements over the last 150 years.

With this approach, the author finds that hurricane activity in the Atlantic has overall increased substantially over the last 150 years, including before the onset of anthropogenic global warming. The results reasonably well mimic the time-varying records of existing storms in recent year, but they produce less storms farther back in time, leading to the overall positive trend. Through some additional analyses the author argues that the large variations in hurricane activity over the last 100 year are due to regional climate variability rather than global warming.

Comments:

The work is interesting and relevant, but there are so many steps and assumptions and adjustments that I do not find the results convincing. First, the downscaling technique only produces the right numbers of storms in the modern era when the number of output storms is rescaled to match the mean value in the last 50 years. So it cannot independently produce the average activity. After this rescaling, it does seem to reproduce the well-known variability over the last 50 years, which is encouraging, but not entirely convincing.

In the second half of the paper, the author provides a very speculative discussion about how assimilating TC data (pressures and locations) might affect the results. However, there is a more pressing issue, which is that there could be biases and trends in the reanalyzed climates due to the fact that the number of surface pressure observations increases with time. If the inherent model climatologies are not the same as the real earth (which is likely), then they would get closer to the real earth climatology over time as more and more data are assimilated. I can see that all three papers documenting the Climate Reanalyses have figures showing an exponential increase in the number of surface pressure observations over the last 150 years.

Some of the statistical analyses leave me with questions. Why are Poisson (exponential) regressions used on some plots, but others use linear? Why are the curves smoothed with a 7-year running mean? Is this to eliminate the multi-annual influences of ENSO, and does that inflate the correlations?

To summarize, I cannot give a strong endorsement of this paper. I think Prof. Emanuel needs to address how changes in the interior atmospheres of the reanalyzed climates may or may not be spuriously changing in ways that affect the TC production by the downscaling scheme, and to have the statistical analyses focused on the raw output with less adjustments.

Here are also some minor comments, by line number:

28: cyclones

39: "no observations of cyclones with non-zero wind speeds" – is a very confusing double negative

73: Here and below I would object to the term "physical" modeling. That means laboratory experiments to me. I would say either "dynamical" or "numerical" modeling.

113: it is

137: linear

In Figure 2, are the top plots landfall frequency or hurricane landfall frequency? The numbers seem too low for the former, but maybe they are reduced due to the smoothing.

292: Reference 23 is not complete.

304: were

355: model

379: What does "held to be" mean here? Does it mean "believed," or fixed to a value?

Reviewer #2 (Remarks to the Author):

Review of "Atlantic Tropical Cyclone Activity Downscaled from Climate Reanalyses Shows Increasing Risk Through the Late 19th and 20th Centuries" by Kerry Emanuel.

This study focuses on an important question of whether long-term changes in Atlantic tropical cyclone activity are evident throughout the known record. Applying a physical-based downscaling approach (random seeding + CHIPS for genesis/intensity + beta-advection for tracks) to three climate reanalyses, author finds although Atlantic tropical cyclone activity might be moderate underestimated in the 19th century, increasing trend is still significant over the past historical period. Using new diagnostic method, author also suggests this increasing tropical cyclone activity is more likely dominated by regional influences in Atlantic sector, rather than global climate change. However, no significant trend of global tropical cyclone activity is detected by the downscaling technique.

Although this study suggests some different results from recent study by Vecchi et al. (2021), both studies highlight the role of regional impacts (e.g., natural climate variabilities; response to aerosols), rather than global anthropogenic-induced climate change influences, on shaping last century Atlantic tropical cyclone activity. This result is of high relevance to the community and the wider field, and the implications for studies involving future climate drivers are well-stated. The proposed downscaling technique is highly reputed and was successfully applied to solve many open questions regarding tropical cyclones and climates. Overall, the manuscript is thoughtfully and logically written. I recommend publication subject to the following minor revisions.

Specific Notes:

1. Figure 4 shows very interesting and impressed result. According to Line 267-269, this is done by decomposing potential intensity into regional and global contributions. However, Reference 23 seems to be unpublished and very limited information can be found. This reviewer will be very appreciated if author can introduce this technique a little more in the Method section. What is perturbation potential intensity?

2. While downscaled Atlantic tropical cyclone activities seem to be better agreed among all three climate reanalyses, why global mean activities differ so much? From eyeballing, CERA20C and

NOAA 20CRv3 are negative correlated even after 1980 (Figure 3a green and red curves). Which dataset is more consistent with IBTrACS? If data assimilation of historical tropical cyclone data may not be attributable, what else source of bias can be? Since CERA20C is coupled while NOAA 20CRv3 is atmosphere-only climate reanalysis data, can air-sea interaction be a important factor influences atmosphere vertical temperature and moisture profiles and thus influences potential intensity and downscaling (as discussed in Section 3.1 of Emanuel 2010 JAMES and Emanuel and Sobel 2013)?

Emanuel, K (2010), Tropical cyclone activity downscaled from NOAA-CIRES reanalysis, 1908–1958. *J. Adv. Model. Earth Syst.* 2, 1–12 (2010).

Emanuel, K., and A. Sobel (2013), Response of tropical sea surface temperature, precipitation, and tropical cyclone-related variables to changes in global and local forcing, *J. Adv. Model. Earth Syst.*, 5, doi:10.1002/jame.20032.

3. Line 104, “coupled-ocean atmosphere model” should be “coupled ocean-atmosphere model”?

4. Line 355, “mode” should be “model”?

Reviewer #3 (Remarks to the Author):

Comments on “Atlantic Tropical Cyclone Activity Downscaled from Climate Reanalyses Shows Increasing Risk Through the Late 19th and 20th Centuries”

This work focuses on understanding Atlantic Tropical Cyclone Activity during the Late 19th and 20th Centuries by applying a downscaling approach to three reanalysis data sets. This work has provided another set of evidence that exhibits the increasing risk of Atlantic tropical cyclone. Overall, this manuscript includes major scientific advancements in quantifying the risk of Atlantic tropical cyclone activity. Before I recommend publication, here are the comments to be considered by the author when revising the manuscript.

Line 13: “analyses” to “reanalyses”

Lines 16-17: Chan et al. (2021) reported that the AMIP-runs based on corrected SST produce no obvious trend since 1885 (its Figure 1). The author could consider adding some discussions to reconcile the results in this study and Chan et al. (2021).

Line 84: change “our” to “my”

Line 85: change “we” to “I”

Line 86: change “our” to “my” or “the”

Line 100: change “we” to “I”

Line 104: change “coupled-ocean atmosphere” to “coupled ocean-atmosphere”

Line 104: change “we” to “I”

Line 113: change “is it” to “it is”

Line 115: “we are” to “I am”

Line 116: change “our” to “my” or “the”

Line 117: change “we” to “I”

Figure 1: Nine legends were included and placed in different positions, but maybe one is enough. The author could consider removing eight legends. The text “Major Hurricanes” were partially covered the figure.

Line 137: “Poisson liner regression” could be changed to “Poisson regression”

Line 148: remove “linear”

Line 153: change “we” to “I”

Line 153: change “we” to “I”

Line 162: change “shows” to “show”

Figure 3: Please add y-axis label. The author may consider removing "linear" in "Poisson linear regressions"

Line 199: change "strong cyclones" to "major hurricanes"

Line 227: change "we" to "I"

Line 228: change "we" to "I"

Lines 230-232: The author could consider discussing Chan et al. (2021).

Line 261: change "NOAA v.2v" to "NOAA v.2c"

Line 269: The author could add more description for the ref 23.

Line 296: change "follows" to "follow"

Line 304: change "we" to "were"

Line 340: change "our" to "my"

Reference:

Chan, D., Vecchi, G. A., Yang, W. & Huybers, P. Improved simulation of 19th- and 20th-century North Atlantic hurricane frequency after correcting historical sea surface temperatures. *Science Advances* (2021).

Atlantic Tropical Cyclones Downscaled from Climate Reanalyses Show Increasing Activity Through the Late 19th and 20th Centuries

(Revised version, 6 October 2021)

Kerry Emanuel¹

Abstract

Historical records of Atlantic hurricane activity, extending back to 1851, show increasing activity over time, but much or all of this trend has been attributed to lack of observations in the early portion of the record. Here we use a tropical cyclone downscaling model driven by three global climate analyses that are based mostly on sea surface temperature and surface pressure data. The results support earlier statistically-based inferences that storms were undercounted in the 19th century, but in contrast to earlier work, show increasing tropical cyclone activity through the period, interrupted by a prominent hurricane drought in the 1970s and 80s that we attribute to anthropogenic aerosols. In agreement with earlier work, we show that most of the variability of North Atlantic tropical cyclone activity over the last century was directly related to regional rather than global climate change. Most metrics of tropical cyclones downscaled over all the tropics show weak and/or insignificant trends over the last century, illustrating the special nature of North Atlantic tropical cyclone climatology.

¹ Lorenz Center, Massachusetts Institute of Technology, 77 Mass. Ave., Rm 54-1814, Cambridge, MA 02139, USA. email: emanuel@mit.edu

Introduction

Changes in the incidence of tropical cyclones are among the most important consequences of climate change, but the short length and often poor quality of historical records of tropical cyclone activity impede detection of climate signals. Here we supplement these historical records with synthetic storms downscaled from global climate analyses.

Only about 12% of the world's tropical cyclones occur in the North Atlantic, but owing to their closer proximity to populated land and to routine airborne reconnaissance since the mid-1940s, they are better observed than tropical cyclones elsewhere¹. Before the era of globally comprehensive satellite coverage, generally regarded as commencing around 1980, observations of tropical cyclones outside the Atlantic were insufficient to achieve complete detection, and estimates of storm intensity were few and subject to large error¹. (An exception was the western North Pacific region, in which some cyclones were surveilled by aircraft from 1945 to 1987.)

Virtually all known reconstructions of tropical cyclone characteristics over the globe have been compiled in a public database known as the International Best Track Archive for Climate Stewardship (IBTrACS)². Outside the Atlantic and western North Pacific, this record is sparse before the satellite era; for example, no tropical cyclone wind speed observations were recorded in the southern hemisphere prior to 1956. In general, the observations become increasingly sparse going back in time.

The IBTrACS North Atlantic tropical cyclone record dates to 1851. Before routine airborne reconnaissance began in the mid-1940s, this record is based on observations from ships and land, including islands. While much of the U.S. eastern seaboard and the Caribbean region was densely populated during this whole period³, both the number and spatial distribution of ship observations evolved in response to changing demand for goods, sizes of ships, and the opening of the Suez Canal in 1869 and the Panama Canal in 1914³. In addition, the increasing use of the "Law of Storms" developed in the early 19th century by Reid⁴, Redfield⁵, and others enabled ships to avoid the cores of tropical cyclones thus affecting observations of the precise location and magnitude of the storms.

Beginning with the work of Vecchi and Knutson⁶ and continuing through a recent paper by Vecchi et al.⁷, attempts have been made to correct for missing tropical cyclones early in the record by estimating how many contemporary (typically post 1971) cyclone tracks would have been observed in the pre-satellite era from land or from ships, with the latter estimates based on the statistics of digitally available ship locations going back to 1851. Such analyses must make a number of assumptions, including thresholds for an observation to count as a tropical cyclone record, the distribution of winds with respect to the storm center, assumptions about whether high winds affecting land would have indeed been recorded, and estimates of whether a succession of reports represent a single storm or multiple storms. The method does not account for possible shifts in the geographic distribution of storms or for changing strategies of ships to avoid the cyclones. The method also privileges the null hypothesis and would diminish the magnitude of any real trend that was present⁶. With these assumptions in mind, this body of research concludes that there are no detectable trends in hurricanes or major hurricanes (Saffir-Simpson category 3 and above) through the entire record, from 1851 to various times in the 21st century.

It is important to distinguish between the lack of an actual trend and the lack of a detectable trend. In essence, the aforementioned work demonstrates that the historical data do not suffice to reject the null hypothesis of no trend in hurricanes or major hurricanes. Any number of violations of the stated assumptions could admit to an actual trend, but there would be no way to detect such a trend using direct historical observations of tropical storms and reasonable, conservative assumptions about the null hypothesis and the detectability of the storms with the observational technologies and platforms of the past.

Here we present results of an alternative approach to estimating past hurricane activity, based on dynamical modeling of tropical cyclones embedded in and driven by coarse-resolution climate as simulated by three global weather forecast models constrained only by surface pressure, sea ice, ocean surface temperature observations, and in one case, marine surface winds. These observations are considered good enough over the past 170 years or so to provide high-quality global climate analyses when assimilated into contemporary numerical weather forecast models⁸⁻¹⁰. The reanalyses also account for changing greenhouse gas concentrations, natural and anthropogenic aerosols, and solar variability. These so-called climate “reanalyses” use advanced numerical models constrained by observations to provide optimal estimates of past climate states. Their horizontal resolution is too coarse to simulate tropical cyclones with any fidelity, but they do assimilate historical records of tropical cyclones, such as they are. We do not here make use of any tropical cyclones explicitly represented in these reanalyses, and indeed their inhomogeneous presence in the reanalyses is an impediment to interpreting the results, as will be discussed later.

As described in greater detail in the Methods section, the downscaling technique operates on the coarse-grained climate reanalysis using monthly mean ocean and atmospheric temperature and atmospheric humidity, and daily winds at two levels in the troposphere. These winds are used to create synthetic time series of winds that are constrained to have the correct monthly means, variances, and co-variances, and to have the observed spectral power distribution.

This space-time distribution of wind, atmospheric temperature and humidity, and sea surface temperature is seeded randomly in space and time with weak proto-tropical-cyclones, which subsequently move according to the synthesized large-scale winds and which grow or decay according to a numerical intensity model driven by the large-scale kinematic and thermodynamic environment. The vast majority of seeds die out quickly, and the small fraction of survivors is regarded as constituting the climatology of these synthesized cyclones. By this means it is possible to generate a very large number of synthetic tropical cyclones, and application of this technique to the recent historical period gives excellent renditions of tropical cyclone climatology^{11,12}.

For the present work, I applied this technique to three analyses: The NOAA 20th Century reanalysis, version 2c⁸ (1851-2014) and version 3¹⁰ (1836-2015), and the European Centre for Medium-Range Forecasts CERA-20C reanalysis⁹ (1901-2010). The CERA-20C reanalysis differs from the NOAA reanalyses in assimilating marine surface winds and assimilating data into a coupled ocean-atmosphere model. I created 100 North Atlantic and 100 global tropical cyclone tracks for each year in each reanalysis, recording the frequency as proportional to the ratio of successful to net (failed plus successful) seeds, as in previously published work^{11,13,14}.

One of the most important differences among these reanalyses is the sea surface temperature (SST) data set assimilated into them. Specifically, prior to 2013, the NOAA v.2c reanalysis

assimilates Sparse Input (SODAsi) version 2 (SODAsi.2¹⁵) with the high latitudes (>60) corrected to COBE-SST2¹⁶. In the NOAA v.3 reanalysis, the SST data were updated to SODAsi.3¹⁷. Although the CERA 20c reanalysis used an interactive ocean, its surface temperature was relaxed back to SST data using the HadISST2 data set¹⁸. Chan et al.¹⁹ demonstrated that different historical gridded SST data sets can appreciably affect explicitly simulated tropical cyclones in global models, and these different SST reconstructions no doubt contribute to the differences in downscaled tropical cyclone activity discussed below.

Here we show that metrics of North Atlantic tropical cyclone activity downscaled from these three climate re-analyses show consistent and substantial increases over the periods of the reanalyses, interrupted by a prominent hurricane drought in the 1970s and 80s, in agreement with uncorrected historical records. Yet downscaled tropical cyclone activity over all the tropics shows no trend in overall frequency but small increases in the more intense storms.

Results

We measure Atlantic tropical cyclone activity by three metrics: The total number of landfalling events, the number of hurricanes in the basin and the number of major hurricanes in the basin. Here we define landfalling events as the passage of the storm center from sea to land, based on a $\frac{1}{4}$ degree bathymetry/topography data set and only include storms whose lifetime maximum intensity exceeded 40 kts (21 ms^{-1}). Because, in the early part of the observational record, it is not always clear whether successive landfalls were by one or more storms, each landfall was here counted as an “event” regardless of whether one or more storms were involved. As this definition is applied to both synthetic and observed storms, the comparison is homogeneous. Note that all landfall events are included, not just landfalls in the continental U.S. During the period considered here, not just the U.S. but much of the Caribbean was densely populated with adequate sources of information about storms³.

The other two metrics are the total number of hurricanes and major hurricanes in the North Atlantic basin, with hurricanes and major hurricanes defined as tropical cyclones with lifetime maximum winds exceeding 63 kts (32 ms^{-1}) and 95 kts (49 ms^{-1}), respectively.

Figure 1 shows the 7-year running means of these quantities downscaled from all three reanalyses together with their historically observed counterparts as recorded in the (uncorrected) IBTrACS data. The choice of 7 years was motivated by a desire to minimize the purely random variability that dominates especially the historical time series at high frequencies, while retaining most of the variability at decadal and longer periods.

The red curves show the downscaled results, which can be compared to the (uncorrected) historically-derived quantities (blue curves). All the downscaled variables have been multiplicatively rescaled to match the mean historical data in the last 50 years of the record. Poisson regression curves are also displayed, and the shading is a measure of the sampling error of the historical data. (Hereafter we use Poisson regression for integer variables and linear regression otherwise.)

All of the regression curves shown in Figure 1 have small p-values, with the largest (0.004) associated with the downscaled landfall counts from the NOAA v.2c reanalysis. This indicates that all of the downscaled Atlantic metrics shown here have statistically significant upward trends over the period, and the smallest trends are those of the tropical cyclones downscaled from the NOAA v.2c reanalysis (the oldest of the three). The pronounced upward trends in these metrics stand in contrast to historical records augmented with estimated missing storms^{6,7,20}.

Storms downscaled from the most recent two reanalyses show very good agreement with the historical data back to 1900, but tropical cyclones downscaled prior to 1900 generally show larger values than those of the historical record, likely reflecting missing storms in the earlier historical record⁷. Poisson regression of the downscaled landfall counts against the historical metrics during the period of mutual overlap of all three reanalysis (1901-2010) yields p-values of 0.09, 3.8×10^{-6} and 2.5×10^{-5} for NOAA v.2c, NOAA v.3, and CERA-20C, respectively.

Downscaled and historical frequencies of tropical cyclones crossing the coastline of the continental U.S. are shown in the top row of Figure 2. As before, only storms whose lifetime maximum wind speed exceeds 40 kts (21 ms^{-1}) are included, but in this case each storm is counted only once.

Both the observed and all of the downscaled U.S. landfall frequencies show upward linear trends that are statistically significant at the $p=0.05$ level. The downscaled frequency trends during the 20th and 21st centuries are small but still significant, yet the observed trend during this period is neither positive nor statistically significant, in agreement with previous work that showed little change in continental U.S. landfall counts^{21,22}.

All three downscalings show statistically significant increases in tropical cyclone power dissipation index at landfall. The power dissipation index is the sum over each year of the cube of the maximum wind speed in each storm at the time of landfall, and it is a measure of the total dissipation of kinetic energy at landfall, not accounting for the effect of storm diameter²³. Power dissipation is also a loose measure of the destructive potential of wind storms²³. These increases are also highly statistically significant over just the 20th and 21st centuries, signifying increasing destructive potential.

Seven-year running means of downscaled tropical cyclone frequencies and major hurricane counts are shown for the whole globe in Figure 3. Here we show the observed quantities from 1980 on, as they are unreliable prior to then. Each downscaled set has been multiplicatively scaled so that the last 40 years match the mean of the observed quantities between 1990 and 2010. As with the Atlantic, only tropical cyclones whose lifetime maximum wind speeds exceed 40 kts (21 ms^{-1}) are included. Tropical cyclone frequencies downscaled from the two NOAA reanalyses show statistically significant trends, one upward and one downward, while those downscaled from CERA-20C show no significant trend. Major hurricanes downscaled from the two NOAA reanalyses show statistically significant upward trends, but most of these trends occurred before 1920 and the major hurricanes downscaled from CERA-20C show no significant trend.

To help understand how much of the variability and trends in these global quantities reflect actual climate signals, a second ensemble member of CERA-20C was also downscaled; the results are shown by the magenta curves in the figures. This captures both the different variability of the global reanalysis climate simulated by the second ensemble member and a different random variability owing to the limited number (100) of tracks per year. Both the

variability and the trends of the second ensemble member differ from the first by order unity, suggesting that neither is statistically robust when the random variability in the reanalyses is accounted for.

The net picture is one of little change over the period. If there are actual climate trends in any of these global metrics, one would have to produce many more tracks per year to reduce the magnitude of the strictly random variability and to run several ensemble members of the reanalyses.

Discussion

Tropical cyclones downscaled from three climate reanalyses show unequivocal increases in Atlantic tropical cyclone activity, in contrast to previously published work based directly on tropical cyclone observations and which attempts to correct for missing storms by sampling contemporary tropical cyclones using statistics of digitally available historical ship routes. These two very different approaches agree, however, that the historical record prior to 1900 likely misses some events, and that there has been little change in continental U.S. tropical cyclone frequency over the 170-year record. The downscaling does suggest, however, that U.S. landfalling storms have become more powerful over time.

Curiously, the same downscaling shows little systematic change in global tropical cyclone metrics. What can account for the large differences between the behavior of downscaled storms in the North Atlantic and elsewhere? Is it possible that defects common to all three reanalyses and/or to the downscaling technique are responsible for these differences?

One possible source of bias in downscaled tropical cyclone trends is the assimilation of historical tropical cyclone tracks into the NOAA reanalyses. (Although, nominally, these data were also assimilated into CERA-20C, in practice many of these observations were rejected by the data assimilation system¹⁰.) As we have discussed, tropical cyclone observations have evolved over time, and it is clear, for example, that there are missing storms in the North Atlantic in the late 19th and perhaps early 20th centuries. Over the rest of the globe, tropical cyclone observations that include intensity data are sparse prior to the era of global satellite coverage that commenced around 1980 (except for the western North Pacific where airborne reconnaissance was conducted from 1945 to 1987). Assimilating tropical cyclone data into these reanalyses potentially introduces artificial trends.

The downscaling model uses only monthly mean thermodynamic inputs on the relatively coarse reanalysis grids, thus the input data have no individual tropical cyclones but may be biased by their collective presence. In general, one may expect the assimilation of tropical cyclones to slightly lower the regional monthly mean surface pressures, and raise the temperature and lower the humidity of the troposphere. (All forms of aggregated convection, while they moisten the immediate vicinity of the aggregates, dry the troposphere as a whole^{24,25}). Lowering ambient surface pressure causes an increase in tropical cyclone potential intensity²⁶, which is one of the main inputs to the downscaling model, but a constant surface pressure was used in calculating potential intensity from the two NOAA reanalyses so in those cases the surface pressure trend would not directly affect the downscaling. Increasing temperature and decreasing humidity in the troposphere would lead to a decrease in potential intensity and would favor decreased genesis frequency; thus, one would expect artificially increasing historical tropical cyclones

assimilated into the reanalysis to cause a spurious negative trend in downscaled storms. Moreover, the rather abrupt increase in assimilated tropical cyclone data for the world outside the North Atlantic at the dawn of the era of satellite observations would produce a comparably abrupt change in downscaled tropical cyclone metrics if indeed the inputs to the downscaling were appreciably affected by the tropical cyclone data. No such abrupt changes around 1980 are apparent in Figure 3. For these reasons, we do not believe that the assimilation of historical tropical cyclone data into the reanalyses is an appreciable source of bias in the downscaled tropical cyclones.

Artificial trends in downscaled tropical cyclone metrics could be induced by biased trends in surface pressure and/or sea surface temperature assimilated into the reanalyses, resulting from the large increase in the number of observations and/or changing instrumentation. With regard to the latter, one would need to explain why such bias trends should be peculiar to the North Atlantic, given that changes in ship instrumentation should have affected observations everywhere. To explore how increasing density of marine observations might have affected the reanalyses, we calculated the surface pressure in the NOAA v.3 reanalyses averaged over each of five tropical cyclone genesis regions and over their respective hurricane seasons. The five basins are the North Atlantic, eastern North Pacific, western North Pacific, North Indian Ocean, and the southern hemisphere. In all of these regions, there are statistically significant upward trends in surface pressure over the period 1836-2015, amounting to 1-2 hPa over the period. The global average surface pressure contributed by all atmospheric constituents except water vapor is held constant in NOAA v.3, and the increase in surface pressure owing to increasing water vapor, assuming fixed relative humidity, is about an-order-of-magnitude less, implying that there is a decrease in surface pressure outside the hurricane genesis regions; this is indeed the case in the NOAA v.3 reanalysis¹⁰.

For the sake of argument, suppose that all of the increase in surface pressure in tropical cyclone genesis regions is an artifact of increasing observations. Further suppose that this increase is compensated hydrostatically by a decrease in tropospheric temperature, constraining the latter to adhere to a moist adiabatic profile. Simple thermodynamics demonstrates that this would lead to an artificial increase in tropical cyclone potential intensity of about $1 \text{ ms}^{-1} \text{ hPa}^{-1}$, thus the $\sim 1.2 \text{ hPa}$ artificial increase in surface pressure in the North Atlantic genesis region would lead to an artificial increase in potential intensity of about 1.2 ms^{-1} , which can be compared to the actual increase of about 6 ms^{-1} in the reanalysis potential intensity in the same region at the same time of year.

To this we add that the increase in genesis region surface pressure in the other tropical cyclone regions, if they are also artificial, would yield about the same artificial increase in potential intensity, yet there are no significant increases in downscaled tropical cyclone activity in these other regions. Another possibility, alluded to earlier, is that the statistical corrections that have been applied to the historical data to account for missing storms have been too large. This could happen, for example, if the various detection thresholds used to determine whether a given storm would have been observed by a given ship or land station were too severe, or if there has been a shift of Atlantic tropical cyclone activity toward regions that were previously more data sparse. Moreover, the method of estimating the number of missing storms privileges the null hypothesis of no trend, leading necessarily to underestimates of any real trends that exist²⁰. It should be remarked that the ship data recorded in the ICOADS data set used in reference⁷ are not equivalent to those used in reconstructing historical hurricane tracks. For the earlier periods,

those who produced the historical tropical cyclone archive used, in addition to ships logs, general news and marine intelligence published in major newspapers²⁷. This may include some reports from ships whose log books have not yet been digitized and are therefore absent from the ICOADS data; indeed, the digitization of older ship observations is a vigorous, ongoing activity with several updates beyond the dataset used by reference⁷ to estimate missing marine observations of tropical cyclones²⁸.

Let us suppose, for the sake of argument, that the downscaling presented here is broadly correct. There are three features of the downscaled Atlantic tropical cyclone climatology shown in Figure 1 that stand out:

1. A substantial upward trend in most metrics
2. A local maximum in most metrics in the 1930s and early 40s
3. A profound depression of activity in the 1970s and 80s
4. A pronounced uptick in activity after 1990

With the exception of the upward trend, these features are also present in the statistically corrected historical record⁷.

What aspect of the physical climate evolution of the North Atlantic region could possibly explain these features, which are not present in the global downscaled tropical cyclone climatology?

One clue comes from a genesis potential index (GPI) calculated over the tropical North Atlantic and over all of the tropics. These indices represent empirically determined relationships between space-time distributions of observed tropical cyclones and monthly mean variables such as potential intensity, ambient low-level vorticity, tropospheric humidity, and wind shear, determined from reanalysis data. We here use a particular GPI²⁹ which has been found to correlate well with rates of genesis of tropical cyclones downscaled using the technique described in this paper. This index is well correlated with rates of genesis of storms downscaled from all three reanalyses, both for the Atlantic and for all of the tropics, with the single exception being the NOAA v.2c for the globe. (For NOAA v.2c, NOAA v.3 and CERA-20C, respectively, the Atlantic correlation coefficients are 0.84, 0.94, and 0.93, and the global coefficients are 0.24, 0.87, and 0.76.)

It is clear from this analysis that the main contribution to increasing Atlantic tropical cyclone activity comes from increasing potential intensity, a measure of the thermodynamic potential for tropical cyclones. (Annual values of late summer Atlantic GPI and potential intensity are correlated with an r^2 of 0.83.) Recently, a technique has been developed to linearly decompose changes in potential intensity into global and regional contributions, with the global contribution assumed to result directly from global climate change³⁰. Figure 4 shows 7-year running means of potential intensity averaged over August-October and over the Atlantic "Main Development Region" (6-18 N, 20-60 W) together with its decomposition into global and regional contributions. The mean over the record has been subtracted from the potential intensity. (The sum of the two terms is very nearly, but not exactly, equal to the full perturbation potential intensity.) These are compared to the annual frequency of tropical cyclones downscaled from the CERA-20C reanalysis. This clearly demonstrates the near total dominance of regional climate

variability over global climate change in controlling downscaled Atlantic tropical cyclone variability, as was also concluded by a recent study⁷. The direct contribution of global climate change to potential intensity is not entirely negligible, with a highly statistically significant increase of about 0.9 ms^{-1} over the period; nor can one rule out an indirect contribution of global change to some of the North Atlantic variability. For example, global warming can lead to changes in the meridional overturning circulation of the ocean, which has a strong effect on North Atlantic climate³¹.

A prominent feature of the observational record of North Atlantic tropical cyclones, storms downscaled from all three reanalyses, as well as tropical Atlantic sea surface temperature and potential intensity, is the pronounced decline from World War II to the late 1970s followed by a rapid rise to the present. This was originally linked to anthropogenic sulfate aerosol emissions by ref. ³² and subsequent modeling work^{33,34} supports such a link, but recent research³⁵ strongly suggests that radiative response to enhanced mineral dust over the tropical North Atlantic, resulting from a sulfate-aerosol-induced drought in the Sahel is responsible for up to half the decline in sea surface temperature and Atlantic hurricanes during this period. The decline and subsequent recovery of Atlantic tropical cyclones closely follow the large post-war increase in sulfate emissions, followed by an equally strong decline as a result of the implementation of clean air regulations. A previously argued role for a naturally occurring multi-decadal climate mode known as the Atlantic Multi-decadal Oscillation³⁶ has been recently cast into doubt³⁷.

Another interesting feature present in all three downscaled Atlantic tropical cyclone records and the historical record is the strong increase in activity during the 1920s. This is also reflected in tropical North Atlantic sea surface temperatures and related quantities such as potential intensity. The increase led to a local peak in activity in the 1930s, corresponding to the North American Dust Bowl period when continental temperatures were also high³⁸.

The centennial changes in North Atlantic regional climate, including tropical cyclone activity, remain an interesting and important topic of research in climate science, and the quality of predictions of future cyclone activity will depend on improved understanding of the causes of recent variability. And, while global measures of tropical cyclone numbers and wind intensity show little change in these results, increasing temperatures strongly imply increasing tropical cyclone rainfall³⁹; a leading cause of damage and mortality from these storms⁴⁰. Moreover, applying the same downscaling technique used here to global climate models indicates substantially increasing tropical cyclone activity in response to global warming^{13,14}, particularly in the northern hemisphere. Clearly the interaction of tropical cyclones with regional and global climate remains a fruitful and important topic of scientific research.

Methods

The technique for downscaling tropical cyclones from coarse-resolution global climate models and reanalyses is described in detail in ref. ^{11,41}. For each year in the dataset, monthly mean sea surface temperature and atmospheric temperature and humidity are used to calculate monthly mean potential intensity and 600 hPa specific humidity and temperature, all of which are inputs to the intensity model described below. The potential intensity was calculated using the algorithm described in ref. ⁴². Daily horizontal winds at 850 and 250 hPa were processed to

calculate the monthly means, variances, and covariances among both components at both levels. These were then used to synthesize time series of winds at both levels, constructed as Fourier series of random phases in time constrained to have the correct monthly means, variances, and covariances and to obey the geostrophic turbulence kinetic energy spectrum in which the kinetic energy per unit wavenumber falls off as the inverse cube of the wavenumber (see supplement to ref. ⁴¹).

The time-evolving climate state is then seeded, randomly in space and time, with weak proto-vortices that serve to initialize the intensity model. The random seeding tapers off toward the equator to prevent storms forming there. The vortices move with a weighted average of synthesized winds at the 250 and 850 hPa levels, according to a “beta-and-advection” model ⁴³. Their intensity is calculated using a deterministic, coupled ocean-atmosphere quasi-balanced axisymmetric numerical model with parameterized effects of environmental wind shear⁴⁴, which is supplied from the synthetic wind time series. The wind and thermodynamic fields are interpolated linearly in time to the given date and time, and bi-linearly in space to the given latitude and longitude of the storm center.

If the intensity model fails to predict an intensity greater than 7 ms^{-1} within two days of initiation, the seed is discarded. In practice, the vast majority of seeds fail this test; only those placed in suitable thermodynamic and kinematic environments thrive. The process is therefore somewhat akin to natural selection. For the present project, we also required the synthesized tropical cyclones to reach a maximum wind speed of at least 40 kts ($\sim 20 \text{ ms}^{-1}$) during their lifetimes. The numbers of both successful and unsuccessful seeds are recorded, and the process continues in a given year until the preset number of successful storms has occurred. The frequency of storms in the given year is taken to be proportional to the ratio of successful to unsuccessful seeds, with the single proportionality constant determined by comparing the time mean to historical observations over the period in which they are deemed reliable

This method has been shown to reproduce many important facets of observed tropical cyclone variability, including histograms of lifetime maximum winds, basin counts, the annual cycle in each basin, the global spatial distribution genesis and track density, interannual variability in the Atlantic, and decadal trends in power dissipation in each basin except the eastern North Pacific¹¹. It also captures the observed poleward migration of the latitude of maximum intensity of tropical cyclones in the western North Pacific⁴⁵, and statistically clusters in a way that is consistent with observations⁴⁶.

Data Availability: The downscaled hurricane track data and data analysis software are freely available for research and education purposes only and may be obtained by contacting the author at emanuel@mit.edu. Recipients will be asked to sign a non-redistribution agreement.

Code Availability: The data contained in all 4 figures as well as MATLAB scripts for plotting the data are freely available at <https://zenodo.org/record/5558992>.

References

1. Landsea, C. W., Harper, B. A., Hoarau, K. & Knaff, J. A. Can we detect trends in extreme tropical cyclones? *Science* **313**, 452–454 (2006).
2. Knapp, K. R., Kruk, M. C., Levinson, D. H., Diamond, H. J. & Neumann, C. J. The International Best Track Archive for Climate Stewardship (IBTrACS): Unifying tropical cyclone best track data. *Bull. Amer. Meteor. Soc.* **91**, 363–376 (2010).
3. Chenoweth, M. & Divine, D. A document-based 318-year record of tropical cyclones in the Lesser Antilles, 1690–2007. *Geochemistry, Geophysics, Geosystems* **9**, (2008).
4. Reid, W. *An attempt to develop the Law of Storms by means of facts, arranged according to place and time; and hence to point out a cause for the variable winds, with a view to practical use in navigation.* (John Weale, 1822).
5. Redfield, W. C. Remarks on the prevailing storms of the Atlantic Coast, of the North American States. *Amer. J. Sci and the Arts* **20**, 17–51 (1831).
6. Vecchi, G. A. & Knutson, T. R. On estimates of historical North Atlantic tropical cyclone activity. *J. Climate* **21**, 3580–3600 (2008).
7. Vecchi, G. A., Landsea, C., Zhang, W., Villarini, G. & Knutson, T. Changes in Atlantic major hurricane frequency since the late-19th century. *Nature Communications* **12**, 4054 (2021).
8. Compo, G. P. *et al.* The Twentieth Century reanalysis project. *Quart. J. Roy. Meteor. Soc.* **137**, 1–28 (2011).
9. Laloyaux, P. *et al.* CERA-20C: A Coupled Reanalysis of the Twentieth Century. *Journal of Advances in Modeling Earth Systems* **10**, 1172–1195 (2018).
10. Slivinski, L. C. *et al.* Towards a more reliable historical reanalysis: Improvements for version 3 of the Twentieth Century Reanalysis system. *Quarterly Journal of the Royal Meteorological Society* **145**, 2876–2908 (2019).
11. Emanuel, K., Sundararajan, R. & Williams, J. Hurricanes and global warming: Results from downscaling IPCC AR4 simulations. *Bull. Amer. Meteor. Soc.* **89**, 347–367 (2008).
12. Emanuel, K. The hurricane-climate connection. *Bull. Amer. Meteor. Soc.* **89**, ES10–ES20 (2008).
13. Emanuel, K. Downscaling CMIP5 climate models shows increased tropical cyclone activity over the 21st century. *Proc. Nat. Acad. Sci.* **110**, 12219–12224 (2013).
14. Emanuel, K. Response of Global Tropical Cyclone Activity to Increasing CO₂: Results from Downscaling CMIP6 Models. *Journal of Climate* **34**, 57–70 (2020).
15. Giese, B. S. *et al.* The 1918/19 El Niño. *Bulletin of the American Meteorological Society* **91**, 177–183 (2010).

16. Hirahara, S., Ishii, M. & Fukuda, Y. Centennial-Scale Sea Surface Temperature Analysis and Its Uncertainty. *Journal of Climate* **27**, 57–75 (2014).
17. Giese, B. S., Seidel, H. F., Compo, G. P. & Sardeshmukh, P. D. An ensemble of ocean reanalyses for 1815–2013 with sparse observational input. *Journal of Geophysical Research: Oceans* **121**, 6891–6910 (2016).
18. Titchner, H. A. & Rayner, N. A. The Met Office Hadley Centre sea ice and sea surface temperature data set, version 2: 1. Sea ice concentrations. *Journal of Geophysical Research: Atmospheres* **119**, 2864–2889 (2014).
19. Chan, D., Vecchi, G. A., Yang, W. & Huybers, P. Improved simulation of 19th- and 20th-century North Atlantic hurricane frequency after correcting historical sea surface temperatures. *Science Advances* **7**, eabg6931 (2021).
20. Vecchi, G. A. & Knutson, T. R. D.-:10. 1175/2010JCLI3810. 1. Estimating annual numbers of Atlantic hurricanes missing from the HURDAT database (1878-1965) using ship track density. *J. Climate* **24**, (2011).
21. Landsea, C. W. Hurricanes and global warming. *Nature* **438**, E11–E12 (2005).
22. Landsea, C. W. Counting Atlantic tropical cyclones back to 1900. *EOS* **88**, 197–200 (2007).
23. Emanuel, K. Increasing destructiveness of tropical cyclones over the past 30 years. *Nature* **436**, 686–688 (2005).
24. Bretherton, C. S., Blossey, P. N. & Khairoutdinov, M. F. An energy-balance analysis of deep convective self-aggregation above uniform SST. *J. Atmos. Sci.* **62**, 4273–4292 (2005).
25. Wing, A. A. & Emanuel, K. A. D.-:10. 1002/2013MS000270. Physical mechanisms controlling self-aggregation of convection in idealized numerical modeling simulations. *J. Adv. Model. Earth Sys.* **6**, 75–90 (2014).
26. Emanuel, K. A. An air-sea interaction theory for tropical cyclones. Part I: Steady state maintenance. *J. Atmos. Sci.* **43**, 585–605 (1986).
27. Fernández-Partagás, J. & Diaz, H. F. Atlantic Hurricanes in the Second Half of the Nineteenth Century. *Bulletin of the American Meteorological Society* **77**, 2899–2906 (1996).
28. Freeman, E. *et al.* ICOADS Release 3.0: a major update to the historical marine climate record. *International Journal of Climatology* **37**, 2211–2232 (2017).
29. Emanuel, K. Tropical cyclone activity downscaled from NOAA-CIRES reanalysis, 1908-1958. *J. Adv. Model. Earth Sys.* **2**, 1–12 (2010).
30. Rousseau-Rizzi, R. & Emanuel, K. A Weak Temperature Gradient Framework to Quantify the Causes of Potential Intensity Variability in the Tropics. *Journal of Climate* 1–48 (2021) doi:10.1175/JCLI-D-21-0139.1.

31. Boers, N. Observation-based early-warning signals for a collapse of the Atlantic Meridional Overturning Circulation. *Nature Climate Change* **11**, 680–688 (2021).
32. Mann, M. E. & Emanuel, K. A. Atlantic hurricane trends linked to climate change. *EOS* **87**, 233–244 (2006).
33. Booth, B. B. B., Dunstone, N. J., Halloran, P. R., Andrews, T. & Bellouin, N. Aerosols implicated as a prime driver of twentieth-century North Atlantic climate variability. *Nature* **484**, 228-U110 (2012).
34. Dunstone, N. J., Smith, D. M., Booth, B. B. B., Hermanson, L. & Eade, R. Anthropogenic aerosol forcing of Atlantic tropical storms. *Nat Geosci* **6**, 534–539 (2013).
35. Rousseau-Rizzi, R. On the climate variability of tropical cyclone potential intensity and Atlantic hurricane activity. Ph.D. thesis, Massachusetts Institute of Technology (2021).
36. Goldenberg, S. B., Landsea, C. W., Mestas-Nuñez, A. M. & Gray, W. M. The recent increase in Atlantic hurricane activity: Causes and implications. *Science* **293**, 474–479 (2001).
37. Mann, M. E., Steinman, B. A., Brouillette, D. J. & Miller, S. K. Multidecadal climate oscillations during the past millennium driven by volcanic forcing. *Science* **371**, 1014–1019 (2021).
38. Cook, B. I., Miller, R. L. & Seager, R. Dust and sea surface temperature forcing of the 1930s “Dust Bowl” drought. *Geophysical Research Letters* **35**, (2008).
39. Knutson, T. R. & Tuleya, R. E. Impact of CO₂ -induced warming on simulated hurricane intensity and precipitation: Sensitivity to the choice of climate model and convective parameterization. *J. Climate* **17**, 3477–3495 (2004).
40. Rappaport, E. N. Loss of Life in the United States Associated with Recent Atlantic Tropical Cyclones. *Bulletin of the American Meteorological Society* **81**, 2065–2074 (2000).
41. Emanuel, K. A., Ravela, S., Vivant, E. & Risi, C. A statistical-deterministic approach to hurricane risk assessment. *Bull. Amer. Meteor. Soc.* **19**, 299–314 (2006).
42. Bister, M. & Emanuel, K. A. Low frequency variability of tropical cyclone potential intensity, 1: Interannual to interdecadal variability. *J. Geophys. Res.* **107**, doi:10.1029/2001JD000776 (2002).
43. Marks, D. G. *The beta and advection model for hurricane track forecasting*. 89 (1992).
44. Emanuel, K., DesAutels, C., Holloway, C. & Korty, R. Environmental control of tropical cyclone intensity. *J. Atmos. Sci.* **61**, 843–858 (2004).
45. Kossin, J. P., Emanuel, K. A. & Camargo, S. J. Past and Projected Changes in Western North Pacific Tropical Cyclone Exposure. *Journal of Climate* **29**, 5725–5739 (2016).
46. Daloz, A. S. *et al.* Cluster Analysis of Downscaled and Explicitly Simulated North Atlantic Tropical Cyclone Tracks. *Journal of Climate* **28**, 1333–1361 (2015).

Acknowledgements: This research was supported by the National Science Foundation under grant ICER-1854929. The author is grateful to Gil Compo of NOAA for his support and advice.

Competing Interests: The author declares no competing interest.

Materials and Correspondence: Kerry Emanuel, Rm. 54-1814 MIT, 77 Mass. Ave., Cambridge, MA 02139, emanuel@mit.edu

Figures

Figure 1: Time series of North Atlantic landfall events (top row), basin hurricanes (middle row) and major hurricanes (bottom row). The landfall frequency includes only storms whose lifetime maximum winds speeds exceed 40 kts (21 ms^{-1}). Red curves show quantities associated with tropical cyclones downscaled from three global climate reanalyses (left to right) and blue curves are derived from uncorrected historical tropical cyclone (IBTrACS) data. All quantities have been smoothed with a 7-year running mean. The downscaled counts have been multiplicatively rescaled to match the mean of the last 50 years of the historical data. The dashed curves show the Poisson regressions of the downscaled and historical data. The shading around the downscaled counts shows the bounds within which 90% of subsamples of the downscaled data lie, with the subsample size equal to the historical counts.

Figure 2: As in Figure 1 but showing the continental U.S. annual landfall frequency (top row) and U.S. landfall power dissipation index (m^3s^{-3} ; bottom row). The landfall frequency includes only storms whose lifetime maximum winds speeds exceed 40 kts (21 ms^{-1}). The linear regression lines in the bottom row are from ordinary linear regression.

Figure 3: Global annual numbers of tropical cyclones (left) and major hurricanes (right) from the IBTrACS historical data (black) and downscaled from the three reanalyses as indicated in the captions. Downsampling from a second ensemble member of CERA-20c is also shown here by the magenta curves, and the historical data are shown by the thicker black curves. Dashed curves show the Poisson regressions for each reanalysis downscaled.

Figure 4: 7-year running means of Atlantic Main Development region (6-18 N, 20-60 W) August-October departures of potential intensity (black, ms^{-1}) from its mean over the record, and its decomposition into global (dashed green) and regional (dashed red) components. The 7-year running mean of the annual frequency of tropical cyclones downscaled from CERA-20C is shown in blue (right axis).

REVIEWERS' COMMENTS

Reviewer #1 (Remarks to the Author):

The author has satisfactorily addressed all of my previous comments.

Here are some small corrections:

line 37: compiled

69: "admit to an actual trend" - use of the word admit is confusing. What are you actually saying?

251: ..everywhere. To...

322: downscaled

391: end with a period

Reviewer #2 (Remarks to the Author):

I appreciate the author for his detailed replies. I don't have additional comments and recommend accepting for the publication.

Reviewer #3 (Remarks to the Author):

The author has fully addressed my comments and I am happy to recommend acceptance at the present form.

Response to Reviewers of the Second Version of
**Atlantic Tropical Cyclones Downscaled from Climate Reanalyses Show Increasing
Activity Through the Late 19th and 20th Centuries**

by Kerry Emanuel

Reviewer 1 cited 5 minor typos. These have all been fixed.